# The Role of TLR-2 in Lethal COVID-19 Disease Involving Medullary and Resident Lung Megakaryocyte Up-Regulation in the Microthrombosis Mechanism

**DOI:** 10.3390/cells13100854

**Published:** 2024-05-17

**Authors:** Giuseppe Pannone, Maria Carmela Pedicillo, Ilenia Sara De Stefano, Francesco Angelillis, Raffaele Barile, Chiara Pannone, Giuliana Villani, Francesco Miele, Maurizio Municinò, Andrea Ronchi, Gaetano Serviddio, Federica Zito Marino, Renato Franco, Tommaso Colangelo, Rosanna Zamparese

**Affiliations:** 1Department of Clinical and Experimental Medicine, University of Foggia, Viale L.Pinto 1, 71122 Foggia, Italy; mariacamela.pedicillo@unifg.it (M.C.P.); ileniadestefano@hotmail.it (I.S.D.S.); francesco_angelillis.546774@unifg.it (F.A.); 2Department of Medical and Surgical Sciences, University of Foggia, Viale L.Pinto 1, 71122 Foggia, Italy; raffaele_barile.552733@unifg.it (R.B.); gaetano.serviddio@unifg.it (G.S.); tommaso.colangelo@unifg.it (T.C.); 3Faculty of Medicine, Università della Campania “Luigi Vanvitelli”, 80131 Naples, Italy; chiara.pannone@studenti.unicampania.it; 4Policlinico Riuniti, University-Hospital, Viale L.Pinto 1, 71122 Foggia, Italy; giuliana-91@hotmail.it; 5Department of Surgery, University of Campania “L Vanvitelli”, 80138 Naples, Italy; dott.miele@gmail.com; 6Forensic Medicine Unit, “S. Giuliano” Hospital, Via Giambattista Basile, 80014 Giugliano in Campania, Italy; maurizio.municino@aslnapoli2nord.it; 7Pathology Unit, Department of Mental and Physical Health and Preventive Medicine, University of Campania “L Vanvitelli”, Via Luciano Armanni, 80138 Naples, Italy; andrea.ronchi@unicampania.it (A.R.); federicazito.marino@libero.it (F.Z.M.); renato.franco@unicampania.it (R.F.); 8Cancer Cell Signalling Unit, Institute for Stem-Cell Biology, Regenerative Medicine and Innovative Therapies (ISBReMIT), IRCCS Fondazione Casa Sollievo della Sofferenza, Viale Cappuccini sc.c., San Giovanni Rotondo, 71013 Foggia, Italy; 9Legal Medicine Unit, Ascoli Piceno Hospital C-G. Mazzoni, Viale Degli Iris 13, 63100 Ascoli Piceno, Italy

**Keywords:** TRLs, TRL-2, SARS-CoV-2, lethal COVID-19, lung disease, ARDS, CD61, micro-thrombosis, megakaryocytes

## Abstract

Patients with COVID-19 have coagulation and platelet disorders, with platelet alterations and thrombocytopenia representing negative prognostic parameters associated with severe forms of the disease and increased lethality. Methods: The aim of this study was to study the expression of platelet glycoprotein IIIa (CD61), playing a critical role in platelet aggregation, together with TRL-2 as a marker of innate immune activation. Results: A total of 25 patients were investigated, with the majority (24/25, 96%) having co-morbidities and dying from a fatal form of SARS-CoV-2(+) infection (COVID-19+), with 13 men and 12 females ranging in age from 45 to 80 years. When compared to a control group of SARS-CoV-2 (−) negative lungs (COVID-19−), TLR-2 expression was up-regulated in a subset of patients with deadly COVID-19 fatal lung illness. The proportion of Spike-1 (+) patients found by PCR and ISH correlates to the proportion of Spike-S1-positive cases as detected by digital pathology examination. Furthermore, CD61 expression was considerably higher in the lungs of deceased patients. In conclusion, we demonstrate that innate immune prolonged hyperactivation is related to platelet/megakaryocyte over-expression in the lung. Conclusions: Microthrombosis in deadly COVID-19+ lung disease is associated with an increase in the number of CD61+ platelets and megakaryocytes in the pulmonary interstitium, as well as their functional activation; this phenomenon is associated with increased expression of innate immunity TLR2+ cells, which binds the SARS-CoV-2 E protein, and significantly with the persistence of the Spike-S1 viral sequence.

## 1. Introduction

Platelets are among the first cells to respond to an infection, monitor the integrity of the endothelium, and initiate the inflammatory-immune and tissue repair processes leading to sterilization of the infectious focus [1]. Patients with COVID-19 have coagulation and platelet disorders, and although they have an average higher platelet count than subjects with forms of Acute Distress Respiratory Syndrome (ARDS) not associated with SARS-CoV-2 [2], in the subgroups that develop thrombocytopenia, the latter represents a negative prognostic parameter being associated with severe forms of the disease and an increase in lethality [3,4].

Platelets express all members of the Toll-Like Receptors (TLRs) family, and their activation has proven to play a relevant role in the amplification of the innate immune response of the host against bacterial and viral infections [5,6].

TLR2 and TLR4 are the main receptors associated with platelet production/function and inflammation [7].

The specific role of TLR-2 in the mechanisms of microvascular thrombosis observed during severe cases of COVID-19 disease.

Important studies in current literature have demonstrated that pathogen recognition by TLR-2 activates Weibel–Palade body exocytosis from endothelial cells [8] and from alpha granules in platelets/megakaryocytes [9,10]. Furthermore, the vWF (Von Willebrand Factor) has a central role in microthrombosis formation during COVID-19 [11].

Megakaryocytes have TLRs, Tumor Necrosis Factor receptors (TNFR1 and 2), receptors for interleukin (IL-1β and IL-6.) These receptors are involved in the activation of the pathway of NF-kB. NF-kB induces both inflammatory and thrombotic responses [12].

Platelets have many TLRs on their cell membrane, including TLR-4. The binding of microorganism (virus, bacteria, fungi, etc.) to TLR-4 induces platelets activation and release of their granule content (many factors involved in different processes) [12].

In this study, we analyzed the immunohistochemical expression of TLR2 and CD61 in the specimens derived from autopsy procedures of patients that died from a critical SARS-CoV-2 infection to potentially associate innate immune hyperinflammation with platelet up-regulation and the microthrombosis mechanism.

## 2. Materials and Methods

### 2.1. Clinicopathological Data of Selected Patients 

In this work, we examined a group of autopsied patients who died as a result of SARS-CoV-2 infection. Autopsy cases were gathered between June 2020 and July 2021, during the outbreak of wild-type SARS-CoV-2 and the early beginnings of the Epsilon and Alpha variants [13].

Patients died after an average of 62.52 days from diagnosis (minimum 20 days, maximum 120 days). The three inclusion criteria are as follows. (1) A positive molecular nasopharyngeal swab was taken prior to or during hospitalization; (2) clinical characteristics were supplied; and (3) histology specimens were available for all studies. The batch contained 12 females and 13 males. The respondents’ ages ranged from 45 to 80. All patients had comorbidities, including arterial hypertension, cardiac hypertrophy, obesity (BMI > 30), type 2 diabetes, obstructive chronic bronchopneumopathy, and chronic nephropathy (see Appendix A). All patients were hospitalized, with lengths of stay ranging from 20 to 122 days.

Further details were published in our recent publication, in which we studied the involvement of TLR-4 in macrophage imbalances in deadly COVID-19 lung illness [14]. The control cases used in this investigation are surgical lung histological preparations from COVID-19-negative individuals.

### 2.2. Autopsy Protocol

Deceased patients were autopsied using a ventilation system with six complete air changes per hour (ACH) in a pressure-negative atmosphere, with air expelled through HEPA filters [Biosafety Level 3 (BSL3)], as per the University of Padova autopsy protocol [15,16].

Lung tissue samples from 25 patients who died from SARS-CoV2 were used as representative examples of COVID-19 fatal ARDS.

### 2.3. Methods

#### 2.3.1. Histological Analysis

Tissues from the lungs of 25 subjects were formalin-fixed and paraffin-embedded (FFPE). All cases were kept in formalin for at least 72 h. The tissues underwent conventional histological examinations. The blocks were divided into 4 μm-thick pieces and stained with hematoxylin and eosin. Two professional pathologists (RF and GP) inspected all histology slides. SARS-CoV-2 RNA was isolated from FFPE tissues using the QIAGEN RNeasy FFPE Kit (catalog no. 73504). The extraction was carried out according to the manufacturer’s instructions. 

As representative examples of COVID-19 fatal ARDS, lung tissue from 25 individuals who died as a result of SARS-CoV2 were examined.

#### 2.3.2. Immunohistochemistry

Immunohistochemical examination was done on formalin-fixed paraffin-embedded specimens from autopsy procedures (COVID-19+ cohort) and standard lung surgical operations (COVID-19− cohort). Single immunostaining was performed using specific monoclonal antibodies against CD61 (clone 2F2) and TLR-2 (rabbit polyclonal, CD-282), as revealed by the standard linked-streptavidin-biotin immunoperoxidase technique (LSAB), developed by 5′ diaminobenzidine and/or alkaline phosphatase methods (Appendix A).

As illustrative cases of COVID-19 fatal ARDS, lung tissue samples from 25 patients who died from SARS-CoV2 infection were used.

In our work, we used CellProfiler 4.2.5 to identify cells that expressed CD61 and TLR-2.

We employed the UnmixColors function to ensure an accurate selection of positive cells based on the intended markers. By employing this function, we were able to discriminate subtle color variations associated with CD61 and TLR-2 expressions, enabling precise segmentation of regions of interest. The program generated signal intensity values ranging from 0 to 100, and based on our initial analyses, we determined that a signal value exceeding 25 serves as a reliable criterion for identifying cells as positive.

This strategy provided us with a robust and reproducible methodology for evaluating CD61 and TLR-2 expressions, significantly contributing to the precision and efficiency of our cellular analyses.

Monoclonal CD61 antibody-stained mature megakaryocytes, with a medium size of 50 µm in bone marrow and 20 µm in lung interstitial tissue, together with pre-megakaryocytes and pro-megakaryocytes with a mean size of 10 µm, and mature platelets with a medium size of 2.65 µm. Different nuclear sizes for pro-megakaryocytes, pre-megakaryocytes, and mature megakaryocytes has been evaluated and reported as large sized multi-lobated and polyploid nuclei of medullary megakaryocytes and small sized and diploid nuclei of interstitial lung megakaryocytes. A positive control slide was used for each immunohistochemical experiment using bone marrow samples.

A series of non-inflamed lungs and chronic interstitial lung pneumonia were used to match samples of SARS-CoV-2-infected hyperinflamed deadly COVID-19 pneumonia. All of the controls have pathological morbidity that requires surgical intervention for therapeutic purposes. In particular, two out of eleven cases had metastasis to the lung from distant carcinomas (one from breast cancer and one from large bowel cancer), six out of eleven had original lung adenocarcinomas, one out of eleven had lung hamartoma, and two out of eleven had complicated lung emphysema. This COVID-19+ cohort with mild chronic lung pneumonia was used to match samples of SARS-CoV-2-infected hyperinflamed deadly instances of pneumonia, resulting in unresolved COVID-19-related ARDS and death, representing the COVID-19+ cohort.

#### 2.3.3. RT-PCR

##### RNA Was Extracted Using 30 μL Buffer and Used for RT-PCR Analysis

The Viral 3 SARS-CoV-2 Kit (BioMol Laboratories srl, Napoli, Italy), which targets the Sarbecovirus envelope gene (E), nucleocapsid (N), and ORF1ab genes of SARS-CoV-2, was used to detect SARS-CoV-2 RNA according to the manufacturer’s instructions. The human RNase P gene was used as a housekeeping gene. The SARS-CoV-2-positive control is a synthetic RNA transcript that includes five SARS-CoV-2 gene targets (E, S, N, ORF1ab and RdRP), as well as the human RNase P gene. An amount of 10 μL of extracted RNA was combined with 5 μL of 4× Real-Time Mix PCR and 5 μL of Primer-Probe Mix. A CFX-96 real-time thermal cycler (Bio-Rad Laboratories, Inc., Hercules, CA, USA) was used for amplification. Settings included one cycle of 2 min at 25 °C, 15 min at 50 °C, and 3 min at 95 °C, followed by 44 cycles of 3 sat 94 °C and 60 sat 60 °C. The spike (S) gene was amplified using the same thermal PCR profile as the Viral 3 SARS-CoV-2 kit, utilizing primers and a probe from Metabion International AG.

#### 2.3.4. In Situ Hybridization (ISH)

We identified Spike and ORF1ab of SARS-CoV-2 in lung tissue using the RNAscope 2.5 high-definition detection kit (Advanced Cell Diagnostics) and two specific probes: ORF1ab (V-nCoV2019-sense-C2) and Spike (V-nCoV2019-S probe). From each block, we obtained sections that were 4 μm-thick. After deparaffinizing the pieces in xylene, they were heated for ten minutes to 95 degrees Celsius. A 10 min hydrogen peroxide incubation was used to block peroxidase activity, which was then permeabilized with protease plus treatment at 40 °C for 30 min. The probe was hybridized for two hours at 40 °C. The signal from the RNAscope was developed with 3,3′-diaminobenzidine, and the nuclei were counterstained with hematoxylin.

A normal autopsy lung tissue sample was placed on each slide as a negative control. All samples were also tested for the presence of well-preserved RNA using the RNAscope^®^ 2.5 LS Positive Control Probe-Hs-PPIB (cat. no. 313908). We defined the following score, which combines staining intensity and diffusion:

Score 0: negative: Indicates no positive staining;

Score 1+: focally positive: Minor and sporadic staining;

Score 2+: diffusely positive: Indicates moderate to intense, diffuse staining.

Evaluation-unfit samples included hemosiderin deposits, indistinguishable dot signals, or endogenous pigments. Two observers who were blinded (ISDS and AR) assessed the results of the ISH test. 

In short, an RNAscope 2.5 High-Definition Detection Kit (Advanced Cell Diagnostics) was used to detect viral sequences Spike (S) of SARS-CoV-2 in lung tissue by ISH detection of the virus. SARS-CoV-2 was found using ISH in the lung tissues of patients with the deadly COVID-19 disease, in accordance with procedures documented in our lab’s earlier work [17,18], where we compared ISH with PCR techniques, and with examples from recent literature [19].

#### 2.3.5. Statistical Analyses

Spearman’s test and ANOVA statistical analysis were utilized to correlate the data to clinicopathological factors. SOFA Statistics 1.4.6, SPSS 28.01.0 Data Analysis and Statistical Software, and Windows Operating Systems were used for data analysis. Non-parametric variables were compared using the Mann–Whitney U-test (Appendix A; Figure 1).

#### 2.3.6. Digital Pathology and Software AI

Sections were digitally scanned with NanoZoomer S60 C13210 series Hamamatsu Photonics K-K, immunostained sections were evaluated with CellSens image analysis software (V1.9^®^, Olympus, Japan, Tokyo) and Visiopharm software version 2021.02 (APP tune, APP Author, Deep Learning with Author AI).

## 3. Results

### 3.1. Main Histopathological Findings

The megakaryocyte lies on the basement membrane of the capillary vessel and its offshoot proplatelets are in contact with two adjacent endothelial cells (Figure 1a).

CD61+ interstitial megakaryocytes line the alveolar space of denuded pneumocytes in a SARS-CoV-2 damaged lung.

In uninfected lung interstitial vessels, there are very rare platelets stained with CD61 marker (Figure 1b).

The interaction in the vessels of SARS-CoV-2 patients of monocytes (CD61−) and platelets (CD61+) is responsible for hypercoagulation, thrombosis, and Disseminated Intravascular Coagulation (DIC)—a late event associated with lethality (Figure 1c–e).

### 3.2. Outcomes of RT-PCR and In Situ Hybridization (ISH)

In our cohort of COVID-19 deadly disease, all patients who died from a lethal type of SARS-CoV-2 infection developed ARDS as a result of lung tissue viral persistence as shown by PCR, as previously reported [20]. Thus, in this investigation, we compared the mean values ± Standard Error Means (SEM) of specific investigated markers (TLR-2, CD61) (Table 1 and Table 2) (Figure 2) to the percentage of Spike-1 (+) cases as assessed by ISH. Table 2 reports the correlation between the ISH results and the RT PCR data.

### 3.3. A Subgroup of Patients with Fatal COVID-19 Lung Disease Had Elevated TLR-2 Expression

The lungs of patients with deadly Covid-19 had significantly greater levels of TLR-2 (Figure 3a–c). Moreover, TLR-2 was upregulated in the lung macrophages of a subset of COVID-19 patients who passed away. Compared to control instances, the average number of TLR-2-stained macrophages per square area is higher; however, because various deceased subjects had varying degrees of TLR-2-mediated hyperinflammation, statistically significant values were not obtained in this cohort. Specifically, using digital pathology-based immunohistochemistry, we found that COVID-19 patients had a mean count of TLR-2-positive macrophages of 128.08 ± SEM 21.789, while the control cases had a mean count of 16.33 ± SEM 5.29.

However, in contrast to the control group of SARS-CoV-2-negative lungs, our analysis revealed a trend toward TLR-2 up-regulation in the lungs of a subgroup of patients with deadly COVID-19. Specifically, we displayed a sub-group (Figure 3d,e and Figure 4) with significant TLR-2 overexpression.

### 3.4. A Subset of Individuals with an Extremely Fatal COVID-19 Lung Illness Had Elevated Levels of Platelet Membrane Glycoprotein IIIa (CD61)

Compared to the SARS-Cov-2-negative lungs in the control group, the lungs of patients with deadly COVID-19 exhibited a substantial up-regulation IN CD61 expression. The Spike-S1-positive cases shown by PCR and ISH methods correlate to a subgroup of patients with considerably elevated numbers of CD61+ platelets and megakaryocytes, linked with a systemic pro-thrombotic state and the production of tissue micro-thrombosis.

Spike-1 (+) status was identified by PCR-based techniques and ISH; means and SEM were acquired by immunohistochemistry expression of CD61 (clone 9C4) (Appendix A, Figure 5, Figure 6 and Figure 7, as assessed by digital pathology analysis.

Using immunohistochemistry, the mean count in SARS-CoV-2 (−) controls was 26.50 ± 7.41 SEM, while the mean count in Spike-1 (−) and Spike-1 (+) COVID-19 deceased patients was 35.23 ± 17.46 SEM and 180.43 ± 35.33 SEM, respectively. The Mann–Whitney U-test indicated the comparisons were statistically significant (*p* < 0.001) (Figure 5 and Figure 6).

### 3.5. Results of RT-PCR and ISH

Endogen control using human RNase P-gene expression (NM-) in the Viral 3 SARS-CoV-2 kit (BioMol Laboratories Srl). The presence of the virus was suggested by the identification of the S gene amplicon. Only 22 out of 25 instances (88%) also had an E gene that was amplified. A total of 24 (96%) of the 25 cases that were studied had an amplified S gene.

The SARS-CoV-2 probe was used in 12 (48%) of the 25 patients that yielded positive results from the ISH test; 8 of these cases had weak and sporadic staining (score 1+), and 4 of these cases had robust and diffuse staining (score 2+).

S ISH staining was commonly found in pneumocytes, hyaline membranes, alveolar macrophages, and air gaps.

A low ISH test sensitivity was indicated by the fact that only 12 out of 25 patients who were positive for S by RT-PCR also exhibited positive staining for spike SARS-CoV-2 ISH. Moreover, no histological damage was observed in this lung tissue when compared to all the other studied patients (Figure 7).

### 3.6. Results Evaluated by Digital Pathology Analysis

CD61 IHC expression was evaluated by digital pathology analysis and showed megakaryocyte up-regulation and diffuse platelet effusion in the lung tissue of lethal COVID-19 disease (Figure 8a).

A control of medullary human megakaryocytes and platelets stained with CD61 monoclonal antibody was evaluated by digital pathology (Figure 8b).

## 4. Discussion

The innate immune response is a key issue to act as the first line of defense against many viral infections [20,21]. The key innate immune sensing receptors are germ line-encoded pattern-recognition receptors (PRRs), which mediate the initial sensing of infection by recognition of pathogen-associated molecular patterns (PAMPs), upon microbial invasion of the host [22]. PRRs belong to different families, including TLRs.

TLR2 and TLR4 are the main receptors associated with platelet production/function and inflammation in the TLR family [23]. It is surprising that TLR2 and TLR4 are not only important components of bacterial pattern recognition but also important receptors regulating megakaryocyte differentiation, as well as platelet production and function.

Infection and lung injuries are associated with the megakaryopoiesis stimulus and increase of lung megakaryocytes. In the lungs of patients who died from COVID-19, megakaryocytes were found in the interstitial alveolar capillaries with the presence of aspects of nuclear hyperchromasia and atypia identified with specific immunohistochemical methods by means of staining positivity for CD61, in association with platelets; platelets within small vessels together with fibrin networks aggregate inflammatory cells including neutrophils and monocyte-macrophages [23].

Lung megakaryocytes in infectious pneumonia generally have an early precursor gene profile compared to mature bone marrow megakaryocytes [24]. Cytokines such as IL-3, IL-6, IL-11, erythropoietin, and thrombopoietin stimulate megakaryocyte progenitor cell maturation [25], while the chemokines CXCL5 (C-X-C motif chemokine ligand 5), CKCL7 (C-X-C motif chemokine ligand 7), inhibit megakaryocyte progenitor cell maturation [26]. In particular, IL-6 together with IL-3 and IL-11, are involved in the activation of endomitosis in megakaryocytes [27] and acts in a TPO-dependent manner together with IL-1β and IL-3, an activated form of tyrosyl-tRNA synthetase (YRSACT), while IL-1α, CXCL5, IGF [insulin-like growth factor]-1) act in a TPO-dependent manner independent.

Thus, we hypothesize that IL-6 upregulation in COVID-19, as well as IL-3 and IL-11 upregulation, is likely involved in the stimulation of megakaryocytes and possibly also resident megakaryocytes in the lung. Platelets and monocytes-macrophages have both been found in the lungs of subjects who died from COVID-19 and it is therefore important to learn more about the interactions between platelets and monocytes-macrophages. Interleukin-6 is an important regulator of thrombopoiesis as megakaryocytes produce IL-6 and express IL-6 on their surface [28]. In vitro and in vivo studies have demonstrated that interleukin 6 is a potent promoter of megakaryocyte maturation [29]. IL-6 is able to increase platelet count thus demonstrating a thrombocytopenic role; however, some functions can be exacerbated in inflammatory pathological conditions as a dysregulation of IL-6 can alter platelet function, making the platelets themselves more sensitive to the action of thrombin and other agonist factors on platelets. IL-6 also increases plasma fibrinogen and von Willebrand factor. These IL-6-mediated modifications of both platelets and the coagulation phase may be decisive in the mechanism of thrombogenesis in the course of pathological inflammation.

Thrombocytopenia has been clearly associated with mortality in patients with COVID-19 [4].

Lethal COVID-19 lung disease is a pathological process with lack of a defense mechanism induced by the virus and a massive innate immunity with hyper-inflammation e necroptosis; in some of the cases, there is a significant number of inflammatory cells, such as in exudative pneumonia, while in other cases, there is a moderate number of inflammatory interstitial cells, the joint participation of pneumocytes, endothelium and maturating megakaryocytes [30]. There are also myeloid-derived cells such as megakaryocytes, platelets and macrophages M1 (CD68+, CD61+), and a modest percentage of lymphocytes. The analyzed cases show an unstopped inflammatory process, an ineffective macrophage M2 response, and a production of cytokines by megakaryocytes/platelets/macrophages/endothelium. In the majority of studied cases, there is a histologically proven early-intermediate diffuse alveolar damage (DAD) phase, with a reduced number than expected showing advanced DAD associated with advanced fibrosis [31].

### Formation of Fibrin-Rich Thrombi Is a Key Aspect of DAD

To date, there is sufficient evidence that lung megakaryocytes are directly infected with SARS-CoV-2 virus. In the bone marrow, viral SARS-CoV-2 N Protein has been detected within megakaryocytes, key cells in platelet production and thrombus formation [32] SARS-CoV-2 interacts with platelets and megakaryocytes via angiotensin-converting enzyme 2 (ACE2)-independent mechanism [33]. However, other authors report that SARS-CoV-2 interacts with platelets and megakaryocytes via angiotensin-converting enzyme 2 (ACE2)-dependent mechanisms [20].

Furthermore, platelets and megakaryocytes are activated directly by inflammatory response indirectly through the cytokine storm [34] and indirectly by the effects on the bone marrow. In particular by the biological effects exerted by IL-6.

In our study, the expression of CD61 showed significant up-regulation in the lungs of patients with lethal COVID-19 when compared to the control group of SARS-CoV-2 negative lungs. The subgroup of patients with significantly high levels of CD61+ platelets and megakaryocytes, associated with a systemic pro-thrombotic state and with the formation of tissue micro-thrombosis, corresponds to the Spike-S1-positive cases demonstrated by PCR and ISH methods.

In this study, the control cases consist of surgical specimens and not autopsy material. The presence of autolytic phenomena present in the cases of deceased patients, absent in controls, represents a limitation of this study.

Platelets have many TLRs on their cell membrane, including TLR4. The binding of the microorganism (virus, bacteria, fungi) to TLR4 induces platelet activation and release of their granule content (i.e., there are many factors involved in different processes) [12].

In our previous study, we demonstrated the upregulation of TLR4 in lethal SARS-COV-2 disease [17]. In this study we demonstrated that the expression of TLR2 similarly was significantly higher in the lungs of patients with lethal COVID-19 when compared to the control group of SARS-CoV-2-negative lungs.

## 5. Conclusions

Microthrombosis in deadly COVID-19 lung disease is associated with an increase in the number of platelets and megakaryocytes in the pulmonary interstitium, as well as their functional activation. It is also associated with an increase in innate immunity TLR2, which binds the SARS-CoV2 E protein, and the persistence of the spike 1 sequence.

## Figures and Tables

**Figure 1 cells-13-00854-f001:**
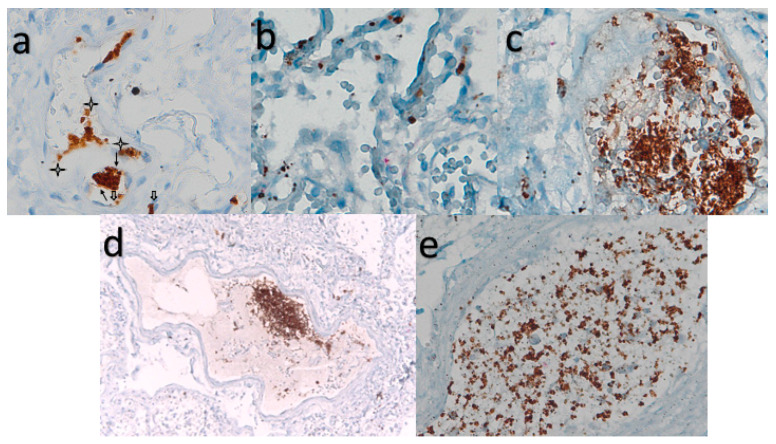
CD-61 immunostaining with standard LSAB-HRP, nuclear counterstaining with Gill’s type-II Haematoxylin; (**a**) Further magnification of an X60 histological picture. Megakaryocyte lies on the basement membrane of the capillary vessel and its offshoot proplatelets are in contact with two adjacent endothelial cells. The staining shows the connections between the megakaryocytes and endothelium (full arrows) and between the endothelium and platelets (empty arrows). Note: the proplatelets (full arrowheads) and released platelets (stars); in the lower-caliber microvessels (<10 micron), red blood cells must stack in a narrow lumen, which in cross-section, is lined by two endothelial cells and one megakaryocyte (bottom of the figure). (**b**) Further magnification of a ×60 histological picture. CD61 in uninfected lung interstitial vessel. (**c**) Further magnification of a ×60 histological picture). Interstitial vessel knots stained with megakaryocytes/platelet CD61 marker in COVID-19 disease. (**d**) Further magnification of a ×20 histological picture. CD61+ platelets in lung micro-thrombus. (**e**) Further magnification of a ×20 histological picture. Monocyte (CD61−) and platelet (CD61+) interaction in vessels of a SARS-COV-2+ patient. This is responsible for hypercoagulation, thrombosis, and Disseminated Intravascular Coagulation (DIC), a late event associated with lethality.

**Figure 2 cells-13-00854-f002:**
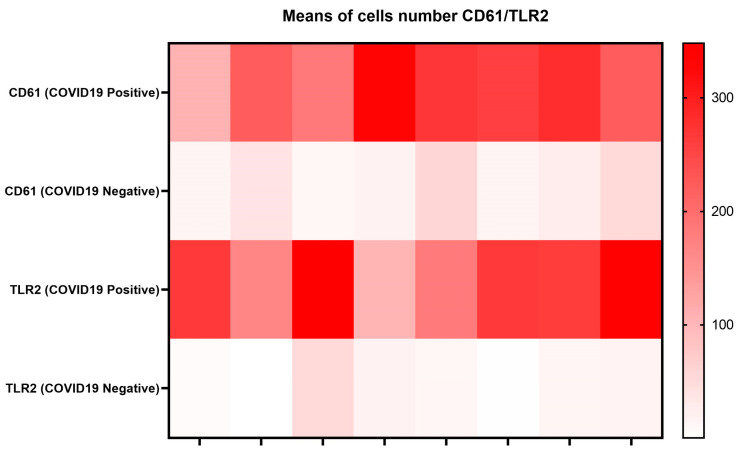
Means of cell numbers CD61/TLR2 COVID-19-positive and COVID-19-negative, as displayed by a heat map.

**Figure 3 cells-13-00854-f003:**
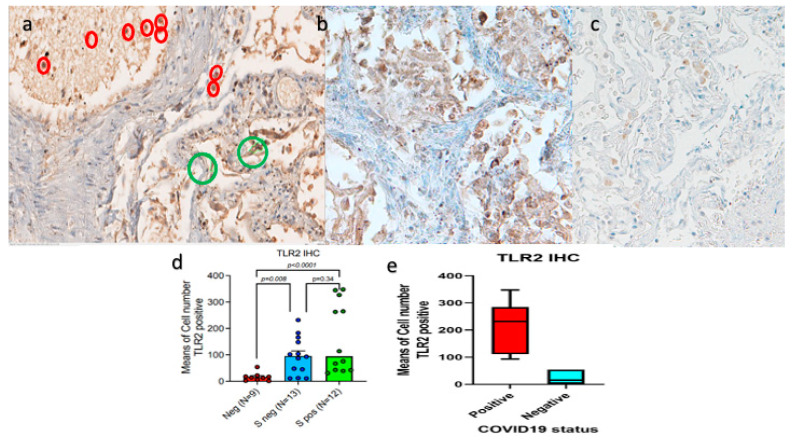
TLR2 IHC expression in megakaryocytes and macrophages. (**a**) Megakaryocytes are highlighted with a green polygonal line, and macrophages with red circles (DAB, Original magnification ×20, NanoZoomer S60 C13210 series Hamamatsu Photonics K-K, Visiopharm software version 2021.02). TLR-2 up-regulation in lethal COVID-19 lung disease (**b**) as compared to uninfected lung and (**c**) further magnification of an x20 histological picture. (**d**,**e**) TLR2 (CD282) was strongly up-regulated in both Spike (+) and Spike (−) SARS-CoV-2-related autoptic lethal lungs when compared to SARS-CoV-2-negative controls; Mann–Whitney U-test statistical test (*p* < 0.001), with a further increase in the levels of expression in Spike (+) cases (*p* < 0.0001).

**Figure 4 cells-13-00854-f004:**
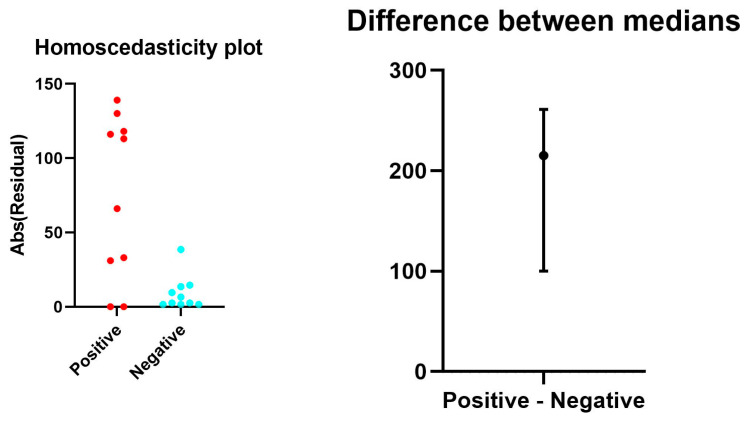
Representative rank plots of TLR2 distribution in the lungs of COVID-19 (+) cases, compared with COVID-19 (−) controls: The ranks plot reveals key differences between COVID-19 patients (red) and controls (blue). COVID-19 patients exhibit a bell-shaped distribution, whereas controls are predominantly distributed toward lower ranks. The trend for COVID-19 patients is slightly positive (increasing rank with value), whereas for controls, it is negative (decreasing rank).

**Figure 5 cells-13-00854-f005:**
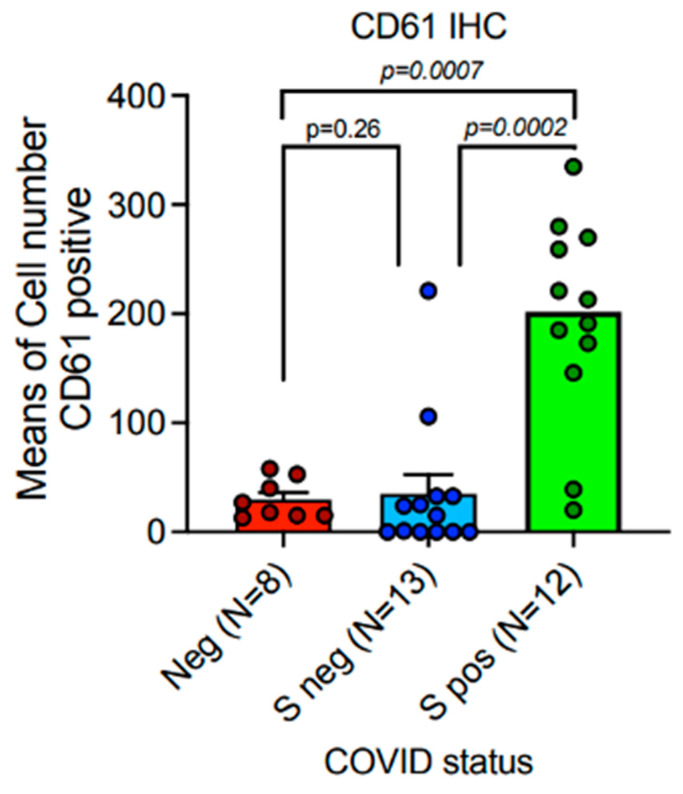
Platelet-megakaryocyte up-regulation is strongly associated with the persistence of Spike-1 in lethal COVID-19 lung disease. Expression of CD61 showed significant up-regulation in the lungs of patients with lethal COVID-19 when compared to the control group of SARS-CoV-2-negative lungs. The subgroup of patients with significantly high levels of CD61+ platelets and megakaryocytes, associated with a systemic pro-thrombotic state and with the formation of tissue micro-thrombosis, corresponds to the Spike-S1-positive cases demonstrated by PCR and ISH methods. Means and standard deviations were obtained by immunohistochemical (IHC) expression of CD61 (glycoprotein IIIa), as evaluated by digital pathology analysis. Spike-positive status was detected by PCR-based methods and in situ hybridization; Mann–Whitney U-test statistical test indicated the comparisons were statistically significant (*p* < 0.001).

**Figure 6 cells-13-00854-f006:**
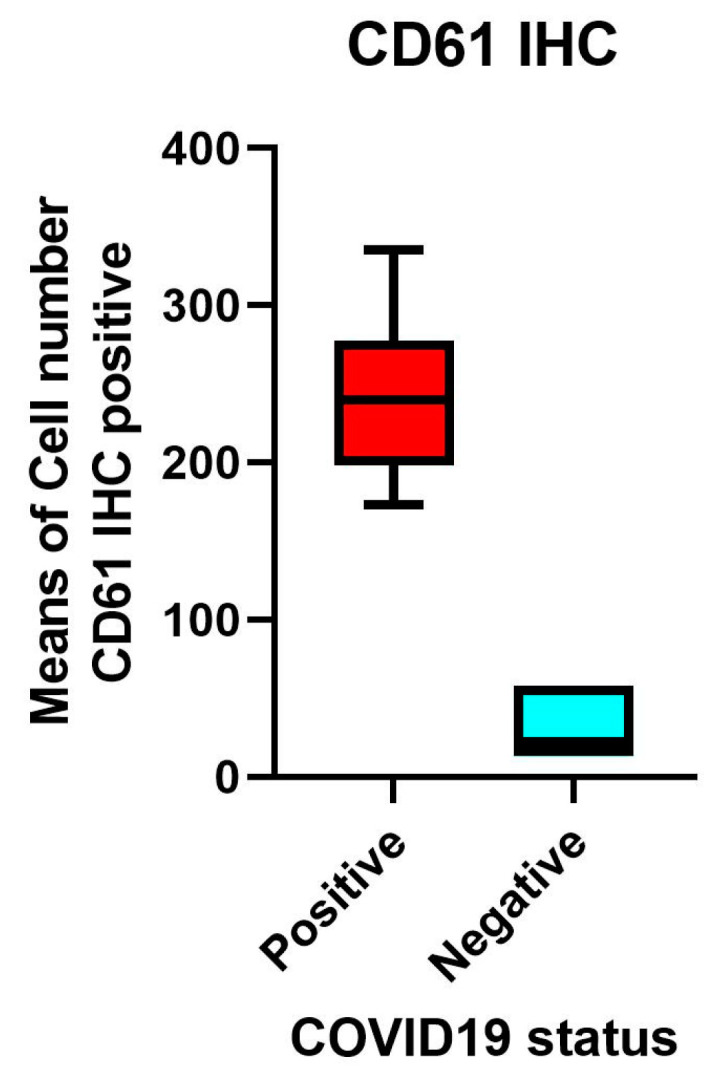
Spike-1 (+) status was detected by ISH. Immunohistochemistry showed a mean count in SARS-CoV-2 (−) controls of 26.50 ± 7.41 SEM, versus mean count of 35.23 ± 17.46 SEM, and 180.43 ± 35.33 SEM in Spike-1 (−) and Spike-1 (+) COVID-19 deceased patients, respectively. Comparisons were statistically significant as evaluated by the Mann–Whitney U-test (*p* < 0.001).

**Figure 7 cells-13-00854-f007:**
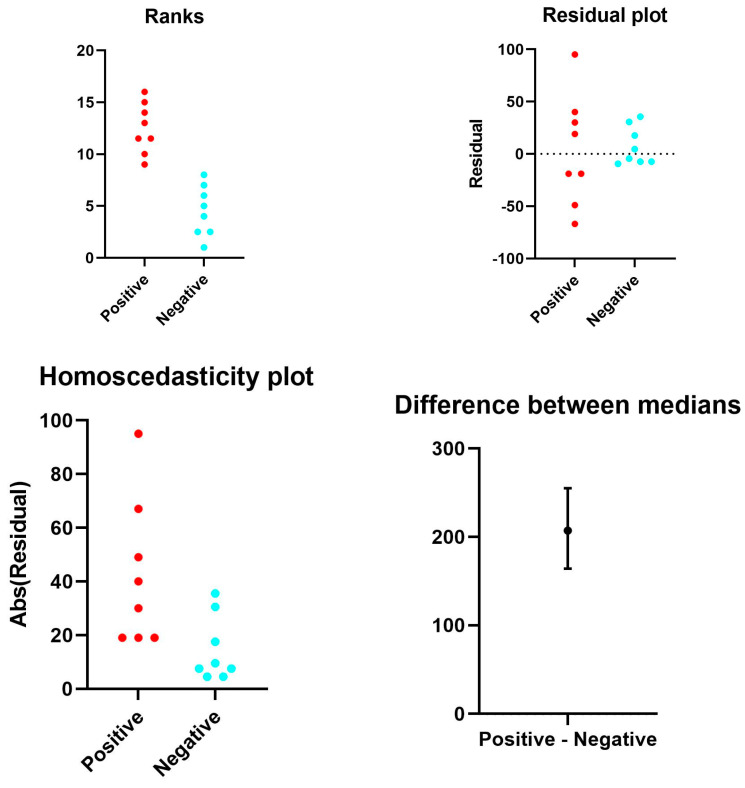
Representative rank plots of CD61 distribution in the lungs of COVID-19 (+) cases, compared with COVID-19 (−) controls: The ranks plot displays two distinct distributions, one for COVID-19-positive cases (red points) and one for control cases (blue points). The positive data exhibits a bell-shaped curve, with the majority concentrated in the central part of the ranks and a slight tendency to increase in rank with value. Conversely, the negative data shows an asymmetric distribution, with the majority accumulated in the lower ranks and a negative slope, indicating a decrease in rank with increasing value. Overall, the negative data appears to be more variable than the positive data, as evidenced by the greater dispersion of the blue points compared to the red ones. Additionally, the plot indicates the presence of more data with positive ranks compared to negative ranks.

**Figure 8 cells-13-00854-f008:**
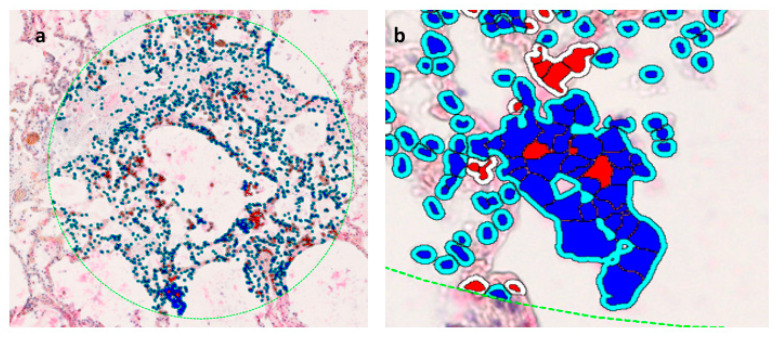
CD61 IHC expression evaluated by digital pathology analysis. (**a**) (Original magnification ×10, NanoZoomer S60 C13210 series Hamamatsu Photonics K-K, Visiopharm software version 2021.02). (**b**) Further amplification from the bottom of the figure (**a**) showing a detail of a megakaryocyte surrounded by platelets. (Original magnification ×60, NanoZoomer S60 C13210 series Hamamatsu Photonics K-K, Visiopharm software version 2021.02).

**Table 1 cells-13-00854-t001:** Summary of Mean ± SEM of TLR2, CD61 immunoreactivity values according to COVID-19 status, and Spike-1 ISH detection.

COVID-19 Status	Immunohistochemistry		Spike-1 (ISH) Positive/Total, (%)
	CD61 Mean ± SEM	TLR2 Mean ± SEM	
Positive	108.43 ± 35.22	163.33 ± 38.50	Spike-1-positive 12/25(48)
	35.23 ± 17.46	95.54 ± 19.66	Spike-1-negative 13/25(52)
Negative	26.50 ± 7.41	16.33 ± 5.29	

**Table 2 cells-13-00854-t002:** Correlation between the ISH results and RT PCR data. ORF1ab refers collectively to two Open Reading Frames conserved in the genomes of coronavirus.

SARS-CoV2 ISH			
Spike +	Spike −	RT-PCR	
12 (48%)	12 (48%)	+	S
0	1 (4%)	−	

## Data Availability

Data are contained within the article.

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
