# Peer review of "The Role of TLR-2 in Lethal COVID-19 Disease Involving Medullary and Resident Lung Megakaryocyte Up-Regulation in the Microthrombosis Mechanism"

_cells, 2024, doi:10.3390/cells13100854_

Round 1

Reviewer 1 Report (Previous Reviewer 1)

Comments and Suggestions for Authors

I was asking the authors to put arrows in Figs 3 and 4 but the revised manuscript did not show the corrections even the authors said they did (only fig 1 has such arrows).

Author Response

Summary

Thank you very much for taking the time to review this manuscript. Please find the detailed responses below and the corresponding correction track changes in the re-submitted files.

Comment. I was asking the authors to put arrows in Figs 3 and 4 but the revised manuscript did not show the corrections even the authors said they did (only fig 1 has such arrows)

 Response 1: We agree with this comment. Therefore, The images have been changed from the first version 

Reviewer 2 Report (Previous Reviewer 2)

Comments and Suggestions for Authors

There are too many figures and many of them should be included as supplementary material. English can be improved and the work should be reviewed by a native speaker; this would make the text more fluid and understandable. There are trivial typing errors, for example affiliation 7 reports VanvItelli which instead must become Vanvitelli as correctly reported in line 18, affiliation 5. The character types must be homogeneous; the font used for affiliations 5 and 6 are different from those used for affiliations 7, 8 and 9. All this is not adequate for the presentation of a scientific work.

The materials and methods are unsatisfactory.

There are several inaccuracies in the materials section; for example, on line 211 the age range of the patients is reported to be 45 to 82 yrs but in the legend of table 1 the range becomes 45 to 80 yrs. The controls are not described in the materials; What are the controls listed in Table 1?  and what are the controls in general? The reviewer imagines that they are deceased COVID-19 negative patients, but this may not be the case. Female control cases are unbalanced; furthermore the sum of the controls is n. 13 but in table 2 they become 11 (negative total n. 11). In the same table the data reported gender, M/F COVID-19 negative do not coincide with those reported in table 1. Therefore the data presented in table 2 do not meet scientific standards.

Methods. The sequence of methods is non-standard, please follow this sequence, histology, immunohistochemistry, RT-PCR, ISH, statistics.

Regarding the immunohistochemical study, the authors declare on line 288-289 that they used autopsy material from deceased patients and routine surgical material as controls. The material is not comparable since autolytic phenomena are generated in autopsy material which do not occur in material derived from surgery. This is a limitation of the study.

In the methods the RT-PCR part should be distinguished from the histopathology part (specific subheading). Furthermore, ISH is covered in two separate methods points; please combine them.

Results. They are difficult to follow and contain inaccuracies. There are too many figures and graphs. The ISH/RT-PCR paragraph, line 439, mainly compares immunohistochemistry and ISH data (Table 4 and Figure 4). The part regarding the comparison between ISH and RT-PCR is shown in Table 5 which is not explained either in the figure legend or in the text. Figure 5 is of no use.

Figure 6, line 493. Legend states that TLR-2 is up-regulated in cases of lethal COVID-19; however, in the next line, 494, reference is made to CD-61 immunostaining which labels platelets / megakaryocytes and leukocytes; this is not evidence of up-regulation of TLR-2.

In figure 7, reference is made to data obtained with the CD-282 antibody (anti-TLR-2), which the previous figure does not show. Furthermore, the acronym CD-282 is the first time it appears in the text and should be included in the immunohistochemistry methods section.

Figure 9 is a complex figure, probably made up of four graphs; the legend does not refer precisely to the individual graphs.

Discussion. Please state the limitations of the study.

Author Response

Response to Review 2

1. Summary

Thank you very much for taking the time to review this manuscript. Please find the detailed responses below and the corresponding corrections track changes in the re-submitted files.

2 Point-by-point response to Comments and Suggestions for Authors

Comments 1: There are too many figures and many of them should be included as supplementary material. English can be improved and the work should be reviewed by a native speaker; this would make the text more fluid and understandable. There are trivial typing errors, for example affiliation 7 reports VanvItelli which instead must become Vanvitelli as correctly reported in line 18, affiliation 5. The character types must be homogeneous; the font used for affiliations 5 and 6 are different from those used for affiliations 7, 8 and 9. All this is not adequate for the presentation of a scientific work.

Response 1: We agree with this comment. Therefore, we changed the figures. we've eliminated some figures and added some lighter ones.We've corrected trivial typos and inserted the same font throughout the text.  

Comments 2: There are several inaccuracies in the materials section; for example, on line 211 the age range of the patients is reported to be 45 to 82 yrs but in the legend of table 1 the range becomes 45 to 80 yrs. The controls are not described in the materials; What are the controls listed in Table 1?  and what are the controls in general? The reviewer imagines that they are deceased COVID-19 negative patients, but this may not be the case. Female control cases are unbalanced; furthermore the sum of the controls is n. 13 but in table 2 they become 11 (negative total n. 11). In the same table the data reported gender, M/F COVID-19 negative do not coincide with those reported in table 1. Therefore the data presented in table 2 do not meet scientific standards.

Response 2: We have, accordingly, revised the materials and methods to emphasize this point.

We have corrected the age range of the patients, to be 45 to 80 on line 103.We have corrected the title of Table 1, which is intended to list only patients who were part of the study. Control cases are listed only in Table 2We've added a reference to control cases (on line 111).The number of control cases is 11 in total, 8 males and 3 females. We have corrected the data in Table 1

Comments 3. Methods. The sequence of methods is non-standard, please follow this sequence, histology, immunohistochemistry, RT-PCR, ISH, statistics.

Regarding the immunohistochemical study, the authors declare on line 288-289 that they used autopsy material from deceased patients and routine surgical material as controls. The material is not comparable since autolytic phenomena are generated in autopsy material which do not occur in material derived from surgery. This is a limitation of the study.

In the methods the RT-PCR part should be distinguished from the histopathology part (specific subheading). Furthermore, ISH is covered in two separate methods points; please combine them.

Response 3. We have, accordingly, revised the methods and we have changed the sequence. We have also supplemented with the data obtained on the usage of digital pathology and AI software.We have distinguished the RT-PCR part from the histopathology part and we have combined the two parts of ISH.In the discussion we have indicated as limitation of this study, the use of autopsy material from deceased patients and routine surgical material as controls 

Comments 4. Results. They are difficult to follow and contain inaccuracies. There are too many figures and graphs. The ISH/RT-PCR paragraph, line 439, mainly compares immunohistochemistry and ISH data (Table 4 and Figure 4). The part regarding the comparison between ISH and RT-PCR is shown in Table 5 which is not explained either in the figure legend or in the text. Figure 5 is of no use.

Figure 6, line 493. Legend states that TLR-2 is up-regulated in cases of lethal COVID-19; however, in the next line, 494, reference is made to CD-61 immunostaining which labels platelets / megakaryocytes and leukocytes; this is not evidence of up-regulation of TLR-2.

In figure 7, reference is made to data obtained with the CD-282 antibody (anti-TLR-2), which the previous figure does not show. Furthermore, the acronym CD-282 is the first time it appears in the text and should be included in the immunohistochemistry methods section.

Figure 9 is a complex figure, probably made up of four graphs; the legend does not refer precisely to the individual graphs.

Response 4: We have, accordingly, revised the results to emphasize this point.

We have replaced figure 5. We have made a reference to Table 5 in the text.

In figure 6 the reference to CD61 was an error that we corrected.

On line 148 we have included the acronym CD-282.

We have also supplemented with the data obtained on the usage of digital pathology and AI software. Comments 5

Discussion. Please state the limitations of the study.

Response 5. In the discussion we have indicated as limitation of this study, the use of autopsy material from deceased patients and routine surgical material as controls.

Round 2

Reviewer 2 Report (Previous Reviewer 2)

Comments and Suggestions for Authors

figures 1, 2, and 3 should be proposed as a single figure with a single legend (a, b, c, d, e)

in figure 5 the green polygonal line and the red circles are not easily visible; please increase the size of the graphic or use an alternative graphic to indicate features

figures 5, 6, 7 and 8 should be proposed as a single figure with a single legend (a, b, c, d, e)

figures 13 and 14 should be proposed as a single figure with a single legend (a) and (b)

move tables 1, 2, and 3 to the supplementary materials

Author Response

We agree with this comment. Therefore, we have edited all images as required

This manuscript is a resubmission of an earlier submission. The following is a list of the peer review reports and author responses from that submission.

Round 1

Reviewer 1 Report

Comments and Suggestions for Authors

This is an interesting study which shows direct interaction between magakaryocytes and monocytes in COVID infected lung. This may answer a mechanism of the fatal coagulation consequence in COVID. The major limitation is the imaging analyses:

1) Fig 3 & 4 need arrows and detail figure legends to orientate reader where are the examples to support their claim. This is particularly important for Fig 4 to show on which type of cells overexpressing TLR2. Maybe a higher magnification insert is necessary to show TLR2 on monocyte and magakaryocyt next to it.

2) The imaging data will be more strengthened if the authors can do a semi-quantitative analyses to show % of magakaryocytes binding to monocytes.

Comments on the Quality of English Language

There are some short paragraph with one or two sentences. This breaks the continuation of reading of the manuscript.

Reviewer 2 Report

Comments and Suggestions for Authors

The Introduction is disorganized and contains a somewhat confusing collection of information. Not really focused on the topic of the paper.should be focused on Toll like receptor 2 given the centrality of this receptor in the title of the work.

The materials and methods are superficial and continually refer to previously published works.

Patient characteristics are very important and these are not described in Table 1 which, as it is proposed, is of no use.

Quantitative evaluation of immunostainings is not detailed.

The immunohistochemical images are of low quality; no scale bar is included in the images. CD61 should also be expressed in endothelial cells but there is no evidence in the provided images.

In the discussion, only occasional reference is made to the data presented in the work. Furthermore, some data present in the discussion are not reported in this work. For example: lines 261-263: In the analyzed cases is shown an unstopped inflammatory process, an ineffective macrophage M2 response and a production of cytokines by megakaryocytes/platelets/macrophages/endothelium.

Very few data to support the manuscript conclusion.

Comments on the Quality of English Language

Several typos and extensive editing of English required.